# A Comparative Study of Several Properties of Plywood Bonded with Virgin and Recycled LDPE Films

**DOI:** 10.3390/ma15144942

**Published:** 2022-07-15

**Authors:** Pavlo Bekhta, Antonio Pizzi, Iryna Kusniak, Nataliya Bekhta, Orest Chernetskyi, Arif Nuryawan

**Affiliations:** 1Department of Wood-Based Composites, Cellulose, and Paper, Ukrainian National Forestry University, 79057 Lviv, Ukraine; kusnyak@nltu.edu.ua; 2LERMAB, Faculte des Sciences, University of Lorraine, Boulevard des Aiguillettes, 54000 Nancy, France; antonio.pizzi@univ-lorraine.fr; 3Department of Design, Ukrainian National Forestry University, 79057 Lviv, Ukraine; n.bekhta@nltu.edu.ua; 4“Shpon Shepetivka” LLC, 30400 Shepetivka, Ukraine; lokiorest@gmail.com; 5Department of Forest Products Technology, Faculty of Forestry, Universitas Sumatera Utara, Medan 20155, North Sumatra, Indonesia; arif5@usu.ac.id

**Keywords:** plastic film-bonded plywood, recycled polyethylene film, formaldehyde release, physical-mechanical properties, bonding strength, wood species

## Abstract

In this work, to better understand the bonding process of plastic plywood panels, the effects of recycled low-density polyethylene (rLDPE) film of three thicknesses (50, 100, and 150 µm) and veneers of four various wood species (beech, birch, hornbeam, and poplar) on the properties of panels were studied. The obtained properties were also compared with the properties of plywood panels bonded by virgin low-density polyethylene (LDPE) film. The results showed that properties of plywood samples bonded with rLDPE and virgin LDPE films differ insignificantly. Samples bonded with rLDPE film demonstrated satisfactory physical and mechanical properties. It was also established that the best mechanical properties of plywood are provided by beech veneer and the lowest by poplar veneer. However, poplar plywood had the best water absorption and swelling thickness, and the bonding strength at the level of birch and hornbeam plywood. The properties of rLDPE-bonded plywood improved with increasing the thickness of the film. The panels bonded with rLDPE film had a close-to-zero formaldehyde content (0.01–0.10 mg/m^2^·h) and reached the super E0 emission class that allows for defining the laboratory-manufactured plastic-bonded plywood as an eco-friendly composite.

## 1. Introduction

Despite the widespread use of traditional wood-based materials such as particle board, oriented strand board (OSB), and medium-density fiberboard (MDF), plywood is still a valuable material used in many industries. In 2020, the industrial production of wood-based panels in the world reached an output of 367 mln. m^3^, including 118 mln. m^3^ (32.2%) of plywood [1]. Synthetic adhesives based on formaldehyde are mainly used for the manufacture of wood-based panels, including plywood [2]. Among the synthetic adhesives, the urea-formaldehyde (UF) adhesives are the most widely used adhesives for the preparation of interior grade composites used for furniture and a wide variety of other applications, accounting for about 85% of the total volume worldwide [3,4].

Along with their many advantages such as chemical versatility, a high reactivity, excellent adhesion properties, solubility in water, relatively low curing temperatures, a short pressing time, ease of transportation, and a relatively low cost [2,5], these adhesives are characterized by certain problems associated with the release of hazardous volatile organic compounds (VOCs), including free formaldehyde, from finished wood composites. Formaldehyde is carcinogenic to humans and harmful to the environment, and its release indoors is associated with adverse human health problems [6,7]. The growing environmental problems and strict legal requirements for free formaldehyde emissions from wood-based panels have posed new challenges to researchers and industry in the development of environmentally friendly engineering wood products with close-to-zero formaldehyde emissions [8]. Therefore, the development of highly efficient ultra-low formaldehyde emissions is necessary for the sustainable production of wood-based panels. The current state of research and recent developments in the field of ultra-low formaldehyde emission wood adhesives and formaldehyde scavengers for manufacturing low-toxic, eco-friendly wood-based panels is summarized in the review [9].

On the other hand, it is well known that environmental pollution and the scarcity of natural resources are becoming pressing issues. Every year, a huge amount of plastic waste is generated worldwide (≈6.3 billion tons) [10]. Only 9% of plastic waste is recycled, another 19% is incinerated, 50% ends up in landfill, and 22% evades waste management systems and goes into uncontrolled dumpsites, is burned in open pits, or ends up in terrestrial or aquatic environments [11]. Plastic waste generated annually per person varies from 221 kg in the United States and 114 kg in European OECD countries to 69 kg, on average, for Japan and Korea [11]. In Ukraine, on average, 250–270 kg of plastic waste is generated annually per person. Moreover, most of the waste goes to landfills, mainly industrial waste, of which only 2–3% is disposed of, others accumulate in landfills and local landfills [12]. Of the total amount of waste generated, almost 50,000 tons is polymer waste, of which low-density polyethylene (LDPE) and high-density polyethylene (HDPE) account for more than 30% of the total amount of polymer waste in Ukraine [13]. In Ukraine, only 10% of waste polymer materials are recycled, while the period of polymers biodegrading, such as plastic bags, is hundreds of years. The combustion of polymeric materials releases hazardous substances that pose a great danger to the environment. That is why the problem of recycling polymer waste is very relevant. Plastic is widely used in many applications, especially in the form of disposable products such as plastic bags, agricultural, and greenhouse films. These wastes mainly consist of polyethylene (PE), polypropylene (PP), polystyrene (PS), and polyvinyl chloride (PVC) [14]. The disposal of plastic is one of the major concerns for the environment due to its slow degradation. Nowadays, recycling waste by using it in production processes has also been in trend because it can prevent environmental pollution and reduce production costs.

Therefore, one of the promising directions for solving the problem of plywood toxicity is the use of thermoplastic polymers, especially recycled, instead of toxic thermosetting adhesives [4,15,16,17,18,19,20]. This not only improves the environmental performance of plywood and its production conditions, which mainly affects the quality and cost of plywood production, but also reduces the negative impact of such polymers on the environment due to the long process of their biodegradation. Recycled thermoplastic films can withstand more than two or three cycles of processing without compromising their physical and mechanical properties [21]. The application of thermoplastic film as an adhesive for the bonding of veneer, apart from the fact that the film is formaldehyde-free, has several other advantages compared with using liquid adhesives, which were described in our previous works [18,19,20]. Moreover, the recycled thermoplastic polymer materials are waste products and are cheaper than virgin polymers [22]. Therefore, the disposal of recycled thermoplastic synthetic waste allows economizing virgin polymer resources and simultaneously protecting the environment.

Nevertheless, recent studies on thermoplastic polymers used as adhesives have mainly focused on the application of virgin polymers for bonding wood veneers in plywood production [16,18,19,20,23]. However, there is already some experience in the use of secondary thermoplastic polymers for bonding veneers. For example, 1.5-mm industrial grade pine and birch veneers were bonded with waste polystyrene (PS) recovered from disposable plates and utensils at the bonding parameters as follows: thermoplastic load 750 g/m^2^, press temperature 200 °C, maximum unit pressure 1.5 MPa, total pressing time 300 s [17]. The authors showed that birch wood having a higher hardness gave higher bonding strengths.

The recycled thermoplastic polymers produced from tetra package waste, domestic film waste, recycled synthetic textile fibers (polyurethane and polyamide-6), and recycled polypropylene are also used as adhesives for bonding the birch wood veneer [15]. It was found that the use of different recycled thermoplastic waste products for the bonding of birch wood veneer guaranteed the shear strength of the materials, which is higher than some virgin polymers and considerably exceeds the adhesive strength of industrial plywood based on phenol–formaldehyde adhesives. The optimal technological parameters for producing samples were noted: pressure 2 MPa, contact time 1–2 min, and a temperature for polyethylene of 130 °C, for polypropylene of 180 °C, and for polyamide-6 of 220 °C.

In another study [24], the recycled 500 g shopping plastic bags, mainly composed of polyethylene, polypropylene, polyvinyl chloride, and polystyrene were used as adhesive for bonding poplar veneer. Before using, the recycled plastic bags were cleaned, processed with a chemical reagent, dried, and shredded. A different amount of recycled plastic bags, 60, 80, 100, and 120 g/m^2^, was spread between veneers. It was concluded that the optimal hot-pressing parameters are as follows: a plastic use of 100 g/m^2^, a hot-pressing temperature of 150 °C, and a hot-pressing time of 6 min. The bonding strength of the plywood decreased markedly in response to the increase in the dosage of plastics.

Other authors [25] made the laminated veneer lumber (LVL), utilizing high-density polyethylene (HDPE) from supermarket plastic bags as an adhesive for bonding wood veneers from amescla wood (*Trattinnickia burseraefolia*) in a laboratory scale. Three HDPE amounts were evaluated: 150 g/m^2^, 250 g/m^2^, and 350 g/m^2^. In general, composite boards showed good quality and mechanical properties were similar or higher than those found in LVL manufactured using thermosetting resin. The composite boards made with the 350 g/m^2^ HDPE amount showed better mechanical and dimensional stability properties. Close contact between the HDPE plastics and the wood cell walls resulted in a stronger physical and mechanical bond [26].

One of co-authors mixed wood flour and low-density polyethylene (LDPE) for rerouting plastics waste to make wood composite plastics (WPC). Results of this study revealed that WPC could be produced with a predominant matrix of LDPE up to 95%. The role of the plastics matrix was beneficial in terms of a shortened degradation in nature, and even the mechanical properties were affected, particularly higher in MOE [27].

Another study [28] demonstrated the suitability of recycling waste milk pouches of 40 and 60 µm thickness made out of low-density polyethylene (LDPE) as the bonding agent in preparing plywood panels from veneers of *Melia dubia* wood. The panels were prepared with varying proportions of LDPE films amounting to a polymer content of 80 g/m^2^, 105 g/m^2^, 210 g/m^2^, and 310 g/m^2^. The polymer content of 210 g/m^2^ was found to be the optimum level to ensure the satisfactory physical and mechanical properties of panels.

It was demonstrated that waste polyethylene in the form of granules between 0.7 mm and 1.3 mm in thickness could be also used in the manufacture of OSB panels, resulting in the enhancement of physical and mechanical properties [29].

In this study, the focus is on recycled LDPE as it is the most widespread type of plastics in the packaging industry (different films) and agriculture (crop protection films, haylage films). Recycled plastics contain a multitude of added chemical additives/contaminants (e.g., pesticide residues, pigments, flame retardants, etc.) [30]. The identification of these chemical additives is quite a difficult concern. It was established that the chemical composition of recycled LDPE was not more complex than that of virgin LDPE [31]. The authors explain this by the fact that in the process of recycling, organic compounds may be partially or selectively removed in the cleaning and/or extrusion steps of the recyclate [31]. Moreover, the recycled LDPE has higher tensile strength and shrinkage, but a lower mass flow index compared to virgin LDPE [32].

However, there is a lack of literature data comparing the adhesive ability of virgin and recycled LDPE polymers. Therefore, the purpose of this study was to obtain a better understanding of the bonding process of plastic plywood with recycled LDPE film when using various wood species and to compare the obtained properties with the properties of plywood panels bonded by virgin LDPE film.

## 2. Materials and Methods

### 2.1. Materials

In the experiments, the rotary-cut veneers of poplar (*Populus alba* L.), birch (*Betula verrucosa* Ehrh.), beech (*Fagus sylvatica* L.), and hornbeam (*Carpinus betulus* L.) with thicknesses of 0.75 mm, 1.55 mm, 0.45 mm, and 1.50 mm, respectively, and with a moisture content of 6 ± 2% were used. The recycled low-density polyethylene (rLDPE) film (LLC “Planet Plastic”, Irpin, Ukraine) with the same dimensions as the veneers and thicknesses of 50 µm, 100 µm, and 150 µm, density of 0.92 g/cm^3^, and melting point of 108 °C was used for the bonding of plywood samples. The amount of plastic rLDPE film at thicknesses of 50 µm, 100 µm, and 150 µm equals 46, 92, and 138 g/m^2^, respectively. Virgin LDPE film with a melting point of 105–110 °C under the same conditions was used for the comparison.

### 2.2. Manufacturing and Testing of Plywood Samples

Three-layer plywood samples were made (Figure 1). Sheet of film was incorporated between the two adjacent veneer sheets. The prepared veneer assemblies were subjected to hot pressing in the lab press at a pressure of 1.4 MPa and temperature of 160 °C for 4.5 min. After hot pressing, the plywood samples were removed from the press and were subjected to the cold pressing at room temperature. The cold pressing was performed to release internal stresses and reduce the warping of samples. Then, the plywood panels were air conditioned at 20 ± 2 °C and 65 ± 5% (RH). Three plywood samples were prepared at each condition.

Density, bending strength (MOR), modulus of elasticity in bending (MOE), shear strength, water absorption (WA), and thickness swelling (TS) of rLDPE film-bonded plywood samples were determined according to the standards [33,34,35,36,37]. The shear strength was measured after pre-treatment for bonding class 1—dry conditions—and plywood test pieces were immersed in water at 20 ± 3 °C for 24 h [35,36]. To determine the WA and TS, the samples were immersed in distilled water for 2 and 24 h according to the EN 317 standard [37]. For each variant, at least ten samples were used for the shear strength test and six samples were used to determine MOR, MOE, WA, and TS.

For each test series, one panel was randomly selected for analysis of formaldehyde content (FC) based on EN ISO 12460-3 standard (gas analysis method) [38]. In addition, urea-formaldehyde (UF) adhesive was used to manufacture plywood samples for the comparison of formaldehyde release from UF and plastic-bonded samples. UF resin with a 67% solid content, Ford cup (4 mm, 20 °C) viscosity of 117 s, spot life of 49 s, and a pH value of 8.2 was used in the experiments. For the preparation of UF adhesive, 20% solution of ammonium chloride as hardener and kaolin as filler were used. The plywood samples using UF adhesive were produced according to the pressing parameters usually used in practice: adhesive spread 110 g/m^2^, pressing temperature, pressure, and time of 160 °C, 1.8 MPa, and 6 min, respectively.

Furthermore, the measurement of the core temperature inside the veneer package under given wood species and thickness of rLDPE film was undertaken. Temperature changes were measured using thermocouples connected to a PT-0102K digital multichannel device [39]. Statistical analysis of the obtained results was conducted using SPSS software program version 22 (IBM Corp., Armonk, NY, USA).

## 3. Results

The properties of rLDPE-bonded plywood samples were compared with the properties of virgin LDPE-bonded samples obtained by us in the previous work [20]. The effect of the veneer wood species on several physical and mechanical properties of plywood samples was found statistically significant. In addition, different thicknesses of the film caused the differences in the properties of samples. The average values of the thickness and density of plywood samples bonded by rLDPE and LDPE films are given in Table 1.

### 3.1. Density of Plywood Samples

Since the veneer of different wood species was used for the manufacture of plywood samples, it was natural that the density of the samples would depend on the wood species. Plywood samples had a lower density using low-density wood species under the same pressing conditions. The lowest average density of 461 kg/m^3^ was recorded for the poplar and the highest of 762 kg/m^3^ for the hornbeam plywood samples bonded with rLDPE film (Figure 2a). The densities of birch and beech plywood samples were 662 and 650 kg/m^3^, respectively, and they differ insignificantly (*p* > 0.05) based on the Duncan’s test. A similar trend in the density values was observed for samples bonded with virgin LDPE film (Figure 2a). The values of the density of the plywood samples were related with the initial density of the veneers. The densities of veneer used in this study were: for poplar wood 390 kg/m^3^, for beech wood 605 kg/m^3^, for birch wood 655 kg/m^3^, and for hornbeam wood 730 kg/m^3^. However, it should be noted that not only the density of the veneer, but also its thickness affects the density of the finished plywood samples. The thickness of the veneer used was different for various wood species (Table 1).

The film thickness has a much smaller effect (*F* = 13.690) on the density of plywood samples compared to wood species (*F* = 803.682) according to the *F*-values of the ANOVA analysis, but this effect was also significant (*p* ≤ 0.05). This can be explained by the much smaller share of film compared to the share of wood in the volume of the plywood sample. As the thickness of the film used increases from 50 to 150 µm, the density of rLDPE and LDPE-bonded plywood samples increases by 4.2% and 4.1%, respectively (Figure 2b). The plywood samples bonded with a film thickness of 50, 100, and 150 µm differ insignificantly with each other in the density values (*p* > 0.05).

A non-strong significant difference in the values of the density of plywood samples bonded with LDPE and rLDPE films was found (*p* = 0.043). Since the films had the same thickness and density, this weak significance can be considered as a consequence of the heterogeneity of veneer among the groups. Wood has a very complex anatomical structure and different properties in various fiber directions. It is very difficult to select veneer sheets identical in structure and properties even within the same wood species. The average density values of the plywood samples that were bonded with LDPE film were higher (656.1 kg/m^3^) than those for samples that was bonded with rLDPE film (629.0 kg/m^3^). Thermosetting adhesives (urea-formaldehyde (UF), melamine-urea-formaldehyde (MUF), and phenol-formaldehyde (PF) also significantly affect the density, MOR, and MOE of plywood panels manufactured with eucalyptus, beech, and hybrid poplar veneers, as evidenced by the results obtained by other authors [40]. On the contrary, Shukla and Kamdem [41] studied laminated veneer lumber (LVL) manufactured from yellow poplar veneers using UF, MUF, melamine formaldehyde (MF), and cross-linked polyvinyl acetate (PVAc), and the results showed no differences.

### 3.2. Bending Strength and Modulus of Elasticity of Plywood Samples

ANOVA analysis showed that both wood species and film thickness and their interaction significantly affect MOR and MOE. The lowest MOR of rLDPE-bonded poplar plywood samples averaged at 61.5 MPa, the highest in hornbeam plywood was 101.6 MPa, and 81.3 and 85.4 MPa in beech and birch plywood, respectively (Figure 3a). There is no significant difference in MOR values between beech and birch plywood samples. There is also an insignificant difference in MOR values among rLDPE-bonded and LDPE-bonded plywood samples. The MOR of LDPE-bonded poplar plywood averaged at 62.5 MPa, 122.8 MPa in hornbeam plywood, and 84.2 and 82.2 MPa in beech and birch plywood, respectively. The average MOR values for the hornbeam plywood were 1.65 times higher than those values for poplar plywood. The density values of hornbeam plywood were also 1.65 times higher than those values of poplar plywood. This indicates a virtually linear dependence of MOR on the density of plywood samples. It is known that density strongly correlates with the strength of wood [42]—the greater the density the greater the strength. This is well confirmed by our research [43,44], and agreed well with the results of other researchers [40,41], who found that the MOR and MOE of plywood panels increase with increasing density.

However, when analyzing the impact of wood species on the MOR and MOE, it should be noted that veneers of different wood species and thicknesses were used in this study. As a result, the plywood samples had different thicknesses (Table 1). It is known that the thickness of the veneer affects the MOR of plywood samples [44]. Several researchers [45,46,47] found that the mechanical strength of plywood panels decreases with increasing veneer thickness due to an average depth of lathe checks. It was established that for a 3 mm thick veneer, the lathe check depth is about 70% of their thickness [48]. The thickness of the plywood panels also affects the MOR and MOE. These dependences are expressed linearly [43,44]—MOR and MOE decrease with increasing plywood sample thickness. This is in good agreement with the data of other authors [46,47], who also found a decrease in bending strength by increasing the thickness of the panel.

As the thickness of the rLDPE film used increases from 50 to 150 µm, the MOR of plywood samples increases by 37.1%, from 71.0 to 97.3 MPa (Figure 3b). For the LDPE-bonded plywood, the MOR of samples increases by 14.6%, from 80.9 to 94.7 MPa. Despite this, the insignificant difference in the values of MOR of the plywood samples bonded with rLDPE and LDPE films of different thicknesses was established. This is in good agreement with previous work [49], which showed a reduction in the properties of plywood panels with low amounts of polymer when using plastic film as an adhesive for bonding. Reducing the polymer content leads to the worse and less complete filling of wood cavities and as a result, fewer adhesive locks are formed, which provide bonding strength. This follows from the concept of mechanical adhesion, since mechanical locking is considered the most likely mechanism for bonding thermoplastic polymers to wood [15,16,18,19,20]. According to Pizzi [50], secondary forces are the dominant mechanism of wood bonding.

The lowest value of the MOE of rLDPE-bonded plywood samples is recorded for the poplar samples of 8772.2 MPa, then for the birch samples of 10,451.6 MPa, for the hornbeam samples of 11,922.7 MPa, and the highest for the beech samples of 13,165.5 MPa (Figure 3c). Thus, the poplar plywood samples had the greatest flexibility and elasticity, and the beech samples, the greatest rigidity. When increasing the film thickness from 50 to 150 µm, and, hence, with an increasing polymer content, the MOE of rLDPE-bonded plywood increases by 15.2% from 10,373.9 MPa for a 50 µm thickness to 11,949.7 MPa for a 150 µm thickness (Figure 3d); plywood becomes stiffer and less elastic. A similar tendency is observed for LDPE-bonded plywood (Figure 3d); under similar conditions, the MOE value increased by 12.8%. The difference between the MOE values for plywood bonded with LDPE and rLDPE films is insignificant (*p* > 0.05) based on the Duncan’s test.

### 3.3. Shear Strength of Plywood Samples

It was established that the shear strength of plywood samples depends significantly on both wood species and film thickness and their interaction. The effect of film thickness was stronger (*F* = 195.981) than the effect of wood species (*F* = 106.074). All plywood samples met the requirements of the EN 314-2 standard [36] with the average values of shear strength (Figure 4a) that were above the limit value (1.0 MPa) indicated in the standard. Beech plywood samples showed the highest shear strength value of 1.77 MPa, and poplar samples showed the lowest shear strength value of 1.08 MPa. Poplar, birch, and hornbeam plywood samples differed insignificantly (*p* > 0.05) in terms of shear strength based on the Duncan’s test. All plywood samples had satisfactory bonding strength for indoor applications. The difference between the values of shear strength for the plywood samples from investigated wood species bonded with LDPE and rLDPE films was insignificant (*p* > 0.05) based on the Duncan’s test.

One of the reasons for the higher bonding strength of beech plywood compared to other wood species used is the use of beech veneer of the smallest thickness (Table 1). Less veneer thickness, fewer lathe cracks in the veneer, and therefore the higher bonding strength. Several authors [51,52] also suggested that the shear strength reflects mainly the quality of veneer. Usually, the thicker the veneer, the deeper and more spaced the lathe checks [46]. The deep lathe checks and rough surfaces in thick veneers significantly reduce the shear strength of the plywood samples. Plywood samples from poplar veneer of lower density and thickness, as well as plywood samples from birch and hornbeam veneers of higher density and thickness, differed insignificantly in terms of bond strength based on the Duncan’s test (Figure 4a). As was described in previous work “the amount of polymer that penetrates per unit volume of poplar veneer with a thickness of 0.75 mm is much greater than the amount of polymer that penetrates per unit volume of birch veneer with a thickness of 1.55 mm or hornbeam veneer with a thickness of 1.50 mm” [20]. In addition, less polymer penetrates the cavities of birch and hornbeam veneer, because their porosity is lower than that of poplar veneer. A linear relationship exists between the bonding strength and the porosity [53].

With an increase in the film thickness from 50 to 150 µm, and, hence, in the polymer content, the shear strength of plywood samples bonded with rLDPE film increases 2.1 times (Figure 4b). Since the bonding strength is ensured by the formation of mechanical locks [15,16], as the film thickness increases, more polymer penetrates the wood cavities and more such mechanical locks are formed. This positively affects the bonding strength. However, it was found that in the case of using a film thickness of 50 µm, the thermoplastic polymer is clearly insufficient to ensure a satisfactory bonding strength. The shear strength of plywood samples bonded with an rLDPE film thickness of 50 µm averaged at 0.77 MPa and was lower than the value of 1.0 MPa specified in standard EN 314-2 [36]. This low bonding strength is due to the low adhesive spread rate for this film thickness, which was equivalent to 46.0 g/m^2^. This is almost three times less than the amount of adhesive used in practice for liquid thermosetting adhesives. On the contrary, a virgin LDPE film thickness of 50 µm provided a bonding strength 1.6 times higher (1.26 MPa) than an rLDPE film of similar thickness (Figure 4b). Whereas, a 150 µm rLDPE film provided a bonding strength (1.60 MPa) 1.2 times higher than an LDPE film of similar thickness. Nevertheless, the average values of shear strength for rLDPE- and LDPE-bonded plywood samples were 1.29 and 1.34 MPa, respectively, and differed insignificantly based on the Duncan’s test.

The interaction of wood species and film thickness also significantly affects the bonding strength. As already mentioned, the bonding strength depends on the quality of the veneer, as well as the degree of penetration of the polymer into the veneer. Wood species with higher porosities provide a better penetration of polymer into the wood. According to several researchers [15,16], in the case of using thermoplastic polymers as an adhesive for the gluing of wood veneer, the concept of mechanical adhesion is suitable. This concept involves good penetration of polymer into the wood and the formation of adhesive locks. In this case, the bonding strength will depend on the number of such locks; the more of them and the deeper they are, the better the bonding strength. Therefore, the spread of polymer on the surfaces and in the structure of wooden elements is of great importance. For the melting of polymer and its spreading on a surface and in the structure of wooden elements, the pressing temperature should be sufficient. As can be seen from the heating curves of the core layer of the veneer package (Figure 5), the accepted pressing temperature was sufficient for this. The thickness of the film or the content of the polymer does not affect the heating rate and it can be neglected. Packages of poplar and beech veneer heat up faster than packages of birch and hornbeam veneer. This is due to both the density of wood and the thickness of the veneer used.

Moreover, unlike other wood composite materials, plywood has a continuous bondline. In the case of a significant porosity of wood or a low viscosity of the polymer and its low consumption, an excessive penetration of the polymer may occur, and it may not be enough to form a continuous bondline of uniform thickness, which leads to a decrease in bonding strength. That is why the film thickness of 50 µm was not sufficient to form a continuous adhesive layer. This caused starvation at the bondline and poor bonding (unsatisfactory bonding strength). Nevertheless, it should be noted that penetration is but one factor contributing to bond performance. The results of several authors [53] verified that compatibilizers and nanoparticles are suitable candidates to improve the bonding strength.

### 3.4. Water Absorption and Thickness Swelling of Plywood Samples

The ANOVA analysis was performed on the data to evaluate the effect of wood species and the thickness of the plastic film on the WA and TS of the plywood samples after 2 and 24 h immersion in water. It was found that both factors affect significantly the WA and TS of samples. The weaker WA (2 h) was observed in birch and hornbeam samples bonded with rLDPE film of 36.1% and 38.4%, respectively (the difference between them is insignificant), the stronger WA (2 h) in beech and poplar samples 46.0% and 48.8%, respectively (the difference between them is insignificant). After soaking in water for 24 h, the lowest WA of 48.7% was observed in birch plywood samples bonded with rLDPE film, and the largest WA of 95.5% in poplar samples (Figure 6a). Although the plywood samples bonded with virgin LDPE film showed a similar trend and less WA values than those bonded with rLDPE film (Figure 6a), but the difference between WA values for these two types of films was insignificant (*p* > 0.05).

The statistical analysis showed that wood species has a stronger effect on TS values of rLDPE-bonded plywood samples for both 2 h and 24 h than the thickness of plastic film. During the first 2 h of soaking in water, the least TS was observed in the poplar plywood samples of 5.0%, and the largest of 10.4% in the hornbeam samples (Figure 6c). Birch, beech, and hornbeam plywood samples did not differ in the values of TS for 2 h based on the Duncan’s test. During the 24 h of soaking in water, the least TS was observed in the poplar plywood samples of 5.7%, and the highest of 11.4% in the hornbeam samples (Figure 6c). Birch and beech plywood samples did not differ in the values of TS for 24 h based on the Duncan’s test. The plywood samples bonded with a virgin LDPE film demonstrated a similar trend in TS values as those bonded with rLDPE film (Figure 6c), and the difference between TS values for these two types of films was insignificant (*p* > 0.05).

As can be seen (Figure 6c), a very small differences between the values of TS for 2 h and 24 h exist for the rLDPE/LDPE-bonded plywood samples from veneer of all investigated wood species. Small differences between the values of WA for 2 h and 24 h were observed only for beech, birch, and hornbeam plywood samples (Figure 6a). This indicates that the samples absorb water and swell most intensively during the first 2 h of soaking in water. For poplar plywood samples bonded with rLDPE film, the WA increases almost twice with the duration of immersion in water from 2 h to 24 h (Figure 6a).

The obtained values of WA and TS are in good agreement with the generally accepted statement that these parameters are strongly related to density [54]. A higher density of plywood samples leads to fewer voids and, consequently, to lower WA, but higher TS. The wood species and the thicknesses of veneer and film affect the density of plywood samples, and therefore their WA and TS. This is well illustrated by the example of poplar plywood samples. A higher porosity (lower density) of poplar veneer compared to other investigated wood species leads to a higher WA of plywood samples bonded with rLDPE and LDPE films (Figure 6a). WA occurs by filling the voids and pores with moisture. On the other hand, for the plywood samples from poplar veneer, the lowest TS was observed (Figure 6c), which is due to the small thickness of the veneer used. As a result, more polymer penetrated per unit volume of this veneer than per unit volume of birch, beech, and hornbeam veneers.

As the film thickness increases, and, hence, the amount of thermoplastic polymer, the WA and TS values of rLDPE-bonded plywood samples decrease for 2 h and 24 h. The lowest values of WA (Figure 6b) and TS (Figure 6d) were observed in the plywood samples bonded with 150 µm thick rLDPE film after 2 h of soaking in water, 34.4% and 7.3%, respectively, and 56.0% and 8.2%, respectively, after 24 h of soaking in water. The 50 μm thick rLDPE film demonstrated the worst WA and TS due to the low polymer content, which was insufficient to fill the wood cavities with the molten polymer.

A similar trend in the influence of wood species and thickness of film on WA and TS was observed for plywood samples bonded with virgin LDPE film. The average values of WA (24 h) for plywood samples bonded with virgin LDPE and rLDPE films were 57.3% and 64.2%, respectively. Based on the Duncan’s test, this difference was significant (*p* ≤ 0.05). On the contrary, the average values of TS (24 h) for samples bonded with virgin LDPE and rLDPE films were 9.3% and 9.6%, respectively, and this difference was insignificant (*p* > 0.05) based on the Duncan’s test.

### 3.5. Pearson Correlation for Properties Versus Density

The obtained results showed that the investigated factors and their interaction affect significantly the properties of plywood samples bonded with rLDPE film. Information on the correlation of the investigated properties with each other and the density, as one of the parameters that can significantly affect the properties of plywood samples, would also be useful. Therefore, Pearson’s correlation coefficient was applied to correlate properties among themselves and with density (Table 2). This coefficient indicates that density correlated significantly with all physical and mechanical properties. The highest correlation was found between density and WA (−0.834). This correlation was moderate and negative, which means that an increase in density causes a decrease in water absorption. The density also correlated well with TS (0.774), MOR (0.583), and MOE (0.617). These correlations were moderate and positive, which means that an increase in density causes an increase in bending strength, the modulus of elasticity, and thickness swelling. No correspondingly useful correlation was observed between density and shear strength (0.111). However, shear strength was positively correlated with MOR (0.498) and MOE (0.728), i.e., an increase in bonding strength means a corresponding increase in MOR and MOE. WA was negatively correlated with MOR (−0.657) and MOE (−0.621), which means that the greater the absorption of water by plywood samples, the lower the values of MOR and MOE.

### 3.6. Formaldehyde Release

Table 3 shows that emissions of formaldehyde from plywood samples made from virgin and recycled LDPE films are very low (0.01–0.10 mg/m^2^·h) compared to conventional UF-bonded (0.66–0.85 mg/m^2^·h) plywood samples. The slight release of formaldehyde can be explained by its presence in natural wood [55]. This result suggests that plastic-bonded plywood reached super-E0 classification and can be considered a product with “zero formaldehyde emissions”. This can be considered a significant contribution from the environmental point of view.

## 4. Conclusions

Currently, eco-sustainability and the reuse of plastic wastes are highly topical issues. This study demonstrated that the rLDPE film is suitable for obtaining environmentally friendly plywood panels from veneers of different wood species with characteristics suitable for use successfully in indoor applications. Wood species and film thickness significantly affect the MOR, MOE, shear strength, and dimensional stability of rLDPE-bonded plywood samples. The use of high-density wood species (beech, birch, and hornbeam) compared to low-density wood species (poplar) resulted in higher MOR, MOE, and TS and lower WA of the samples. No clear effect of the density of wood species on the shear strength was found. With an increase in the thickness of plastic film, and, hence, in the polymer content, the values of MOR, MOE, and shear strength of rLDPE-bonded plywood samples increase by 37.1%, 15.2%, and 107.7%, respectively, whereas the values of WA and TS decrease by 25.6 % and 26.9%, respectively. Furthermore, the physical (except WA) and mechanical properties showed no statistical differences between the rLDPE-bonded and virgin LDPE-bonded plywood samples. The main advantage of these polymers is that the amount of formaldehyde emission from the plastic-bonded plywood is close-to-zero (0.01–0.10 mg/m^2^·h). These results confirm the effectiveness and the environmental benefits in the use of the recycled plastic wastes.

## Figures and Tables

**Figure 1 materials-15-04942-f001:**
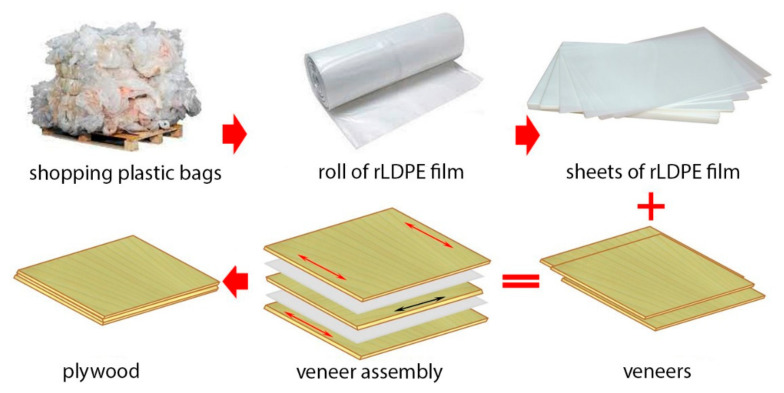
Schematic of plywood samples production.

**Figure 2 materials-15-04942-f002:**
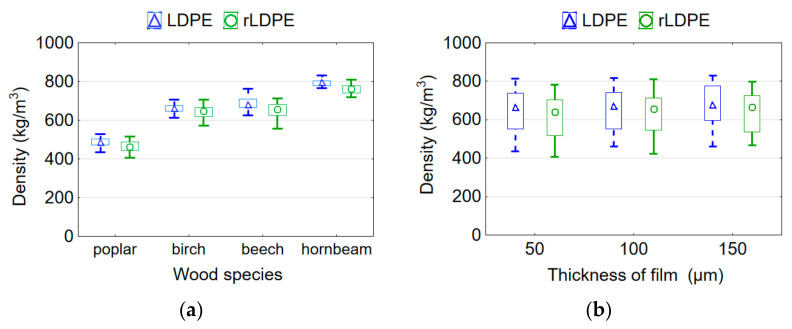
Mean plots of density of plywood samples depending on wood species (**a**) and thickness of rLDPE/LDPE film (**b**).

**Figure 3 materials-15-04942-f003:**
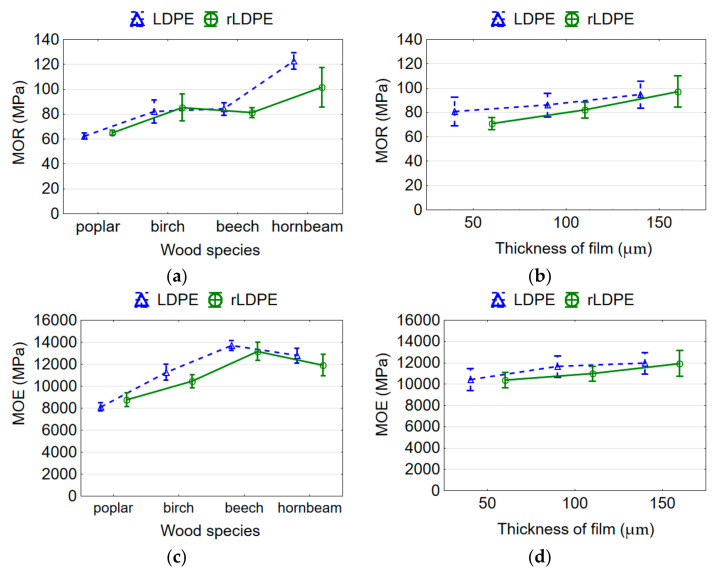
Mean plots of bending strength (**a**,**b**) and modulus of elasticity (**c**,**d**) of plywood samples depending on type of adhesive, wood species, and thickness of rLDPE/LDPE film.

**Figure 4 materials-15-04942-f004:**
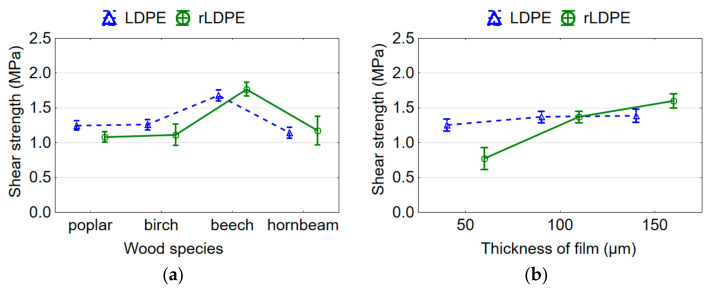
Mean plots of shear strength of plywood samples depending on type of adhesive, wood species (**a**) and thickness of rLDPE film (**b**).

**Figure 5 materials-15-04942-f005:**
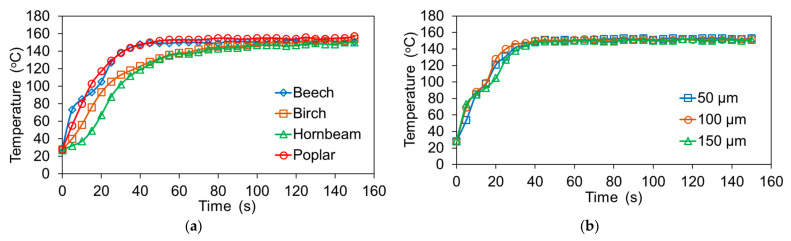
Core temperature curves of plywood samples made with (**a**) veneers of different wood species and rLDPE film of 150 µm thickness, and (**b**) beech veneers and different thicknesses of rLDPE film.

**Figure 6 materials-15-04942-f006:**
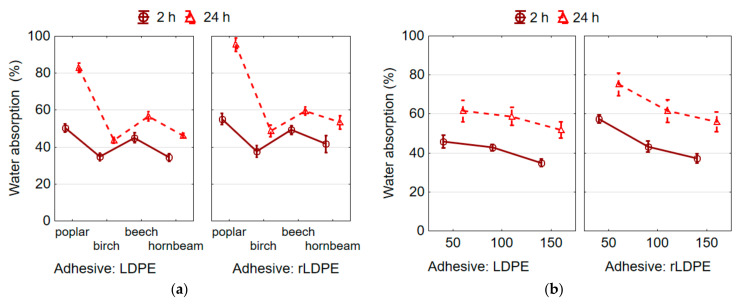
Mean plots of WA (**a**,**b**) and TS (**c**,**d**) of plywood samples depending on wood species (**a**,**c**) and thickness of film (**b**,**d**).

**Table 1 materials-15-04942-t001:** Thickness and density of plywood samples.

Wood Species	Thickness of Veneer (mm)	Thickness of Film (µm)	Thickness of Plywood Samples (mm)		Density of Plywood Samples (kg/m^3^)	
			rLDPE	LDPE	rLDPE	LDPE
Poplar	0.75	50	2.19 (0.02) *	2.19 (0.03)	435.36 (19.94)	463.86 (14.48)
		100	2.23 (0.06)	2.23 (0.04)	463.91 (22.45)	488.74 (15.80)
		150	2.22 (0.04)	2.23 (0.03)	484.48 (13.87)	505.11 (23.46)
Beech	0.45	50	1.39 (0.07)	1.32 (0.05)	618.25 (25.36)	661.16 (17.68)
		100	1.31 (0.03)	1.35 (0.02)	659.83 (14.94)	677.45 (19.69)
		150	1.42 (0.09)	1.39 (0.06)	673.05 (39.87)	720.20 (27.75)
Birch	1.55	50	4.57 (0.09)	4.41 (0.11)	649.93 (13.65)	668.21 (18.90)
		100	4.62 (0.07)	4.52 (0.11)	644.95 (36.16)	651.82 (43.34)
		150	4.69 (0.09)	4.58 (0.06)	632.36 (39.23)	659.53 (13.06)
Hornbeam	1.50	50	4.47 (0.09)	4.35 (0.13)	749.41 (17.66)	784.63 (15.08)
		100	4.49 (0.16)	4.18 (0.03)	765.83 (34.76)	790.09 (15.82)
		150	4.33 (0.13)	4.52 (0.03)	771.01 (15.96)	802.13 (10.94)

* Values in parenthesis are standard deviations.

**Table 2 materials-15-04942-t002:** Pearson correlation (*r*) for properties versus density of plywood samples.

Property	Density	WA (24 h)	TS (24 h)	Shear Strength	MOR	MOE
Density	1					
WA (24 h)	−0.834 ^a^	1				
TS (24 h)	0.774 ^a^	−0.554 ^b^	1			
Shear strength	0.111	−0.334	−0.287	1		
MOR	0.583 ^b^	−0.657 ^b^	0.113	0.498 ^b^	1	
MOE	0.617 ^b^	−0.621 ^b^	0.325	0.728 ^a^	0.704 ^a^	1

^a^ Correlation is significant at the 0.01 level; ^b^ Correlation is significant at the 0.05 level.

**Table 3 materials-15-04942-t003:** Formaldehyde release of plywood samples.

Formaldehyde Release (mg/m^2^·h)				
Adhesive	Wood Species			
	Beech	Birch	Hornbeam	Poplar
UF	0.85 (0.03)	0.69 (0.02)	0.73 (0.03)	0.66 (0.03)
LDPE—50	0.03 (0.004)	0.06 (0.001)	0.10 (0.002)	-
LDPE—100	-	0.08 (0.001)	-	-
LDPE—150	-	0.04 (0.002)	0.06 (0.002)	0.07 (0.002)
rLDPE—50	-	0.05 (0.002)	-	0.09 (0.001)
rLDPE—100	-	-	0.10 (0.002)	0.03 (0.001)
rLDPE—150	0.06 (0.002)	0.01 (0.001)	0.10 (0.002)	-

Values in parentheses represent standard deviations.

## Data Availability

The data that support the findings of this study are available upon reasonable request from the authors.

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
