# Peer review of "A Comparative Study of Several Properties of Plywood Bonded with Virgin and Recycled LDPE Films"

_materials, 2022, doi:10.3390/ma15144942_

Round 1

Reviewer 1 Report

Owing to environmental concerns there is a great interest to substitute low-density polyehylene  (LDPE)  to plywood traditional adhesives. The paper scope is to evaluate if recycled low-density polyehylene films may efficiently substitute the virgin one to obtain, by this way, a much more eco-friendly material.

At moment I cannot express a favorable opinion towards publication for the reasons listed below:

-          The English should be revised.

-           Taking into account the scope of the paper  the recycled low-density polyethylene used should be better defined: which is the source, previous history, information about the possible degradation  state  and possible reduction of properties of interest….

-          Details of the experimental (materials and characterization techniques) must be reported in the “experimental” section . The reader must not be obliged to read them in another manuscript so as indicated on  lines 151-152. In the experimental section also the plywood construction details are to be given. In particular the source of the veneers and their thickness must be given altogether with all other available  information on their properties.

-          When a figure consists of several plots I suggest to distinguish them  with the aid of a letter (Fig 1a and 1b…); this should be exploited when discussing the results

-          All the studied properties are reported as a function of the veneer type and of the plywood thickness. Please clarify which is the plywood thickness of the samples in the first case.   In the second case I understand that the reported values are the medium  of all the veneer type ones. The result would be the same if the plots are repeated for each veneer type? Alternatively the dependences are always  the same for whatever veneer type? Please clarify and specify better in the paper

-          Have the authors measured the density of LDPE and  rLDPE? (see line 206) Which are the values? Why the heterogeneity of the veneer should systematically make the LDPE playwood density be greater? (so as reported on  lines 206-210). Please clarify

-          On lines 244 the authors state: “MOR and MOE decrease with increasing plywood sample thickness.” This is the opposite of the trends shown in Fig. 3 . In fact plywood sample thickness shouldn’t increase with film thickness? Please clarify

-          On lines 257-258 the authors say: “Despite this, the insignificant difference among these films has been established regarding the influence of their thickness on the MOR of samples”. I understand they are comparing LDPE and  rLDPE plywood results. Please clarify and specify better

-          How were chosen the durations (2 and 24h) of the water absorption and thickness swelling tests? Is 24h long enough to reach the maximum absorption?

-          I think that  lines 343-394 are to be deeply revised. I find many  statements questionable and/or very  criptic. Some statements are clearly wrong like the one on lines 346-347: “Whereas the WA for 24 hours is already more strongly influence had the wood species than the thickness of the film.” I suggest to clarify, at the discussion beginning,  which are the factors the authors consider may affect  the results. Moreover the WA and TS after 2h shoudn’t depend on water adsorption kinetic?  The same after 24h if this is not a saturation time! Please clarify

Reviewer 2 Report

The article submitted by Bekhta et al. (materials-1797312), entitled “A Comparative Study of Several Properties of Plywood Bonded With Virgin and Recycled LDPE Films”, investigated the effects of recycled LDPE film on the properties of plywood panels. They found that there was on significant difference in the properties between plywood bonded with virgin and cycled LDPE films, which suggests that the use of the recycled LDPE film as a n adhesive for thermoplastic-bonded plywood seems to be feasible. The article can help the reader understand the situation of replacing the glue with the thermoplastic films. However, the bonding mechanism between the wood and the LDPE film, virgin and recycled, lacked and the article were basically phenomenologically from the experiments, for the benefit of the reader, some concerns are suggested to be considered:

1)     The properties of the LDPE-bonded plywood, included virgin and recycled film, were investigated, and compared carefully in this study. However, the reason behind them was still unclear. Especially, the interface structure between the wood and LDPE film should be investigated carefully in this study.

2)     The recycled LDPE film should be introduced and evaluated carefully, since it can help the reader understand the situation of the used materials. Is there any difference between the virgin and recycled LDPE films? Do they shrinkage after heated?

3)     It is interesting to use thermoplastic film as an adhesive. However, the bonding effects can be influenced by the processes and/or the fabrication method. About this point, an article was recommended, Polymer Composites, 2020, 41, 60-72, although the used processing technology were different. Actually, fused deposition molding is another typical application of plastics bonding. Authors may mention it and give a simple introduction about this concern.

4)     In table 1, does the zero exist before the point? The numbers are somewhat strange.

In conclusion, some revisions are suggested before its publication.

Round 2

Reviewer 1 Report

The issues raised were quite  satisfactorily affressed

Reviewer 2 Report

The revised version is better than before. I recommend its publication in present form.